# A Novel Morphological Parameter Predicting Fibrotic Evolution in Myeloproliferative Neoplasms: New Evidence and Molecular Insights

**DOI:** 10.3390/ijms23147872

**Published:** 2022-07-17

**Authors:** Vincenzo Fiorentino, Pietro Tralongo, Maurizio Martini, Silvia Betti, Elena Rossi, Francesco Pierconti, Valerio De Stefano, Luigi Maria Larocca

**Affiliations:** 1Dipartimento di Scienze Della Salute Della Donna, del Bambino e di Sanità Pubblica, Fondazione Policlinico Universitario Agostino Gemelli IRCCS, 00168 Rome, Italy; vincenzo.fiorentino@guest.policlinicogemelli.it (V.F.); pietrotralongo@gmail.com (P.T.); francesco.pierconti@unicatt.it (F.P.); 2Dipartimento di Patologia Umana Dell’adulto e Dell’età Evolutiva, Università Degli Studi di Messina, 98125 Messina, Italy; maurizio.martini@unime.it; 3Dipartimento di Diagnostica per Immagini, Radioterapia Oncologica ed Ematologia, Fondazione Policlinico Universitario Agostino Gemelli IRCCS, 00168 Rome, Italy; silvia.betti@unicatt.it (S.B.); elena.rossi@unicatt.it (E.R.); valerio.destefano@unicatt.it (V.D.S.); 4Unicamillus, International Medical University in Rome, 00131 Rome, Italy

**Keywords:** myeloproliferative neoplasms, polycythemia vera, idiopathic myelofibrosis, megakaryocytes

## Abstract

Philadelphia-negative chronic myeloproliferative neoplasms (MPNs) represent a group of hematological disorders that are traditionally considered as indistinct slow progressing conditions; still, a subset of cases shows a rapid evolution towards myelofibrotic bone marrow failure. Specific abnormalities in the megakaryocyte lineage seem to play a central role in this evolution, especially in the bone marrow fibrosis but also in the induction of myeloproliferation. In this review, we analyze the current knowledge of prognostic factors of MPNs related to their evolution to myelofibrotic bone marrow failure. Moreover, we focused the role of the megakaryocytic lineage in the various stages of MPNs, with updated examples of MPNs in vitro and in vivo models and new therapeutic implications.

## 1. Introduction

The term Philadelphia-negative chronic myeloproliferative neoplasms (MPNs) refer to a heterogeneous group of hematological disorders which originate from the neoplastic transformation of a pluripotent stem cell and are associated with myeloproliferation, extramedullary hematopoiesis, splenomegaly and, in due course, bone marrow fibrosis (MF). According to the WHO 2016 classification, MPNs can be divided into Polycythemia Vera (PV), Essential Trombocythemia (ET) and idiopathic (primary) Myelofibrosis (IMF) in the prefibrotic and overt form [1]. For over two decades, MPNs have been considered as indistinctly slow-progressing conditions [2,3]. However, recent clinical evidence highlighted a subset of cases [4] with a rapid evolution towards myelofibrotic bone marrow failure, placing interest in developing personalized prognosticators and timely therapeutic strategies against this evolution, correlating latest advances in MPNs’ molecular profiling with differences in clinical outcomes [5,6].

In fact, alongside MPNs’ driver gene mutations (JAK2, CALR, MPL), molecular profiling identified other gene mutations, involving for example DNA methylation (TET2, DNMT3A, IDH1/2), histone modification (ASXL1, EZH2), RNA splicing (U2AF1, SRSF2, SF3B1), DNA repair (TP53) and signal transduction (NRAS, CBL). These mutations can coexist with or without driver gene mutations, affecting the evolution and prognosis of MPNs [5,6].

On this basis, different groups developed several prognostic scores, mainly based on clinical, laboratory and molecular parameters, with less emphasis on morphological and immunophenotypic data [7]. Given the improvements and advances in MPN molecular profiling, the newer models included JAK2, CALR and MPL mutation status in addition to the IPSS parameters, so that the prognostic prediction in IMF patients can be improved [4]. Furthermore, novel insights were supported by a deep analysis of genomic subsets with different clinical prognoses [5]. Recent publications have introduced new prognostic models for PMF, respectively MIPSS70 (mutation-enhanced international prognostic scoring system for transplant-age patients) [6], MIPSS70+ version 2.0 (karyotype-enhanced MIPSS70) and GIPSS (genetically-inspired prognostic scoring system) [8,9]. As the previous models, other ones have been recently introduced for both ET and PV, namely MIPSS-ET and MIPSS-PV, underlining the prognostic importance of spliceosome gene mutations [10].

In opposition to this, all these predictive models do not consider other parameters as morphological or phenotypical features, with the exception of BM fibrosis grade in the MIPPS70 model (Table 1).

### 1.1. MPNs’ Molecular Landscape, In Vivo and In Vitro Models and Possible Novel Therapeutic Strategies

Many of the discoveries on the pathogenesis of MPNs are due to in vivo and in vitro models that have made it possible to reproduce this type of pathology more and more faithfully. There are several animal models of myeloproliferative neoplasms, used to investigate the role of mutations in the development of MPNs, or the impact of additional factors in MPN phenotype modulation. These models are schematically represented in Figure 1 (in vivo models) and in Table 2 (in vitro models).

### 1.2. GATA-1 Low Models

The thrombopoietin-treated (TPO-high) model and the GATA-1 low model are two murine models of MPN used to evaluate the megakaryocyte lineage in the MPNs pathogenesis and evolution [11,14]. The first model develops a myeloproliferative disorder mimicking human myelofibrosis, characterized by leukocytosis, anemia, thrombocytosis, splenomegaly, extramedullary hematopoiesis, fibrosis and osteosclerosis. This model is very useful for evaluating some pathogenetic mechanisms associated with fibrosis development, such as the role of transforming growth factor-beta1 (TGF-β1). The second model consists of the virtual abolishment of GATA-1 expression in megakaryocytes, while the protein continues to be expressed in erythroid cells, although at significantly lower levels [14]. These mice develop a progressive myeloproliferative disorder that has many features of myelofibrosis after 1 year of life and reduced levels of GATA-1 have also been demonstrated in the megakaryocytes of patients with IMF [14,15]. Moreover, mice with a MK-specific deficiency of the transcription factor-encoding gene GATA1 show elevated numbers of immature MK in the BM.

### 1.3. Lysyl Oxidases Models

It was also demonstrated that GATA1 (low) mutation was associated with low ploidy megakaryocytes with an extensive matrix of fibers due to the overexpression of lysis oxidase (LOX) [16]. Lysyl oxidases (LOXs) have been demonstrated to be important in this process by cross-linking collagens and elastins through deamination of lysins and hydroxylysins, resulting in a stiffer extracellular matrix (ECM) consistency [17]. Lysyl oxidases are expressed in immature megakaryocytes and downregulated in mature megakaryocytes, but upregulated in MF patient megakaryocytes and in murine models of MF [16,18]. Lysyl oxidase inhibition has shown efficacy in Gata1low and JAK2V617F mouse models of MF [19,20]. However, a novel phase 2 study of simtuzumab, a monoclonal inhibitor of LOX2, did not reduce bone fibrosis in patients with MF [21]. It was also demonstrated that the inhibition of LOX via the administration of β-aminopropionitrile could stop the progression of the myelofibrosis [22]. This last model is particularly used as a preclinical model for drug testing.

### 1.4. Profibrotic Agent Models

In addition, MK from individuals with MPN, in particular with IMF, secrete increased levels of the fibrotic cytokines such as TGF-β, compared to MK from healthy individuals, and the ECM microenvironment, especially the fibronectin component, is able to sustain progenitor cell proliferation and megakaryopoiesis in a TPO-independent manner.

These pro-fibrotic cytokines would presumably act mainly in the microenvironment near to those MK clusters, which are, in turn, their main producers. Furthermore, the criteria defining the megakaryocytic activation could represent the morphological counterpart of what is postulated by in vitro and in vivo studies regarding the role of MK in the BM fibrotic evolution of patients with MPN.

In this respect, our recent study demonstrated that Megakaryocytic Activation (M-ACT), a new morphological parameter defined by the coexistence of emperipolesis in MKs, MK clustering and peri-MK fibrosis in bone marrow biopsies (BMB) at diagnosis, could represent the morphological counterpart of what is postulated by in vitro and in vivo studies regarding the role of megakaryocytes in the bone marrow fibrotic evolution of patients with MPNs. This parameter represents a consistent and early predictive marker, to be settled at ‘time zero’ of diagnosis and to be further prospectively analyzed for its potential to hasten a closer follow-up and to target MK-dependent fibrotic evolution [7].

Moreover, we identified an additional morphological parameter, defined as Megakaryocentric Fibrosis (MKF), which could be helpful in the diagnostic phase of MPNs. We defined this parameter as a peculiar distribution pattern of collagen fibers. MKF is the production of a collagenic fibrotic reinforcement at the center of megakaryocytic aggregates, which results in being—so to speak—“tattooed”. The deposition of collagen fibers, although focal, becomes more marked, with thick shoots, bridging and a singular arrangement entirely surrounding the megakaryocytes. This particular pattern of distribution of collagen fibers, on the basis of our preliminary data, seems to identify a subset of MPNs with a more aggressive course and is, therefore, an equivalent of M-ACT even in the absence of megakaryocytic emperipolesis and clustering.

As a matter of fact, Cerquozzi S. and Tefferi A. [23] and Malara A. et al. [24] showed how patients with MPNs and fibrotic evolution displayed a considerably increased count of bone marrow MKs with an abnormal nuclear/cytoplasmic ratio, a reduced polyploid state and a tendency to cluster. Moreover, GATA-1 low mice showed numerous immature bone marrow MKs with unusual neutrophil emperipolesis, which could account for sustaining MF by releasing fibrogenic MK cytokines and neutrophil proteases in the microenvironment of in vivo experiments [15,25].

**Figure 1 ijms-23-07872-f001:**
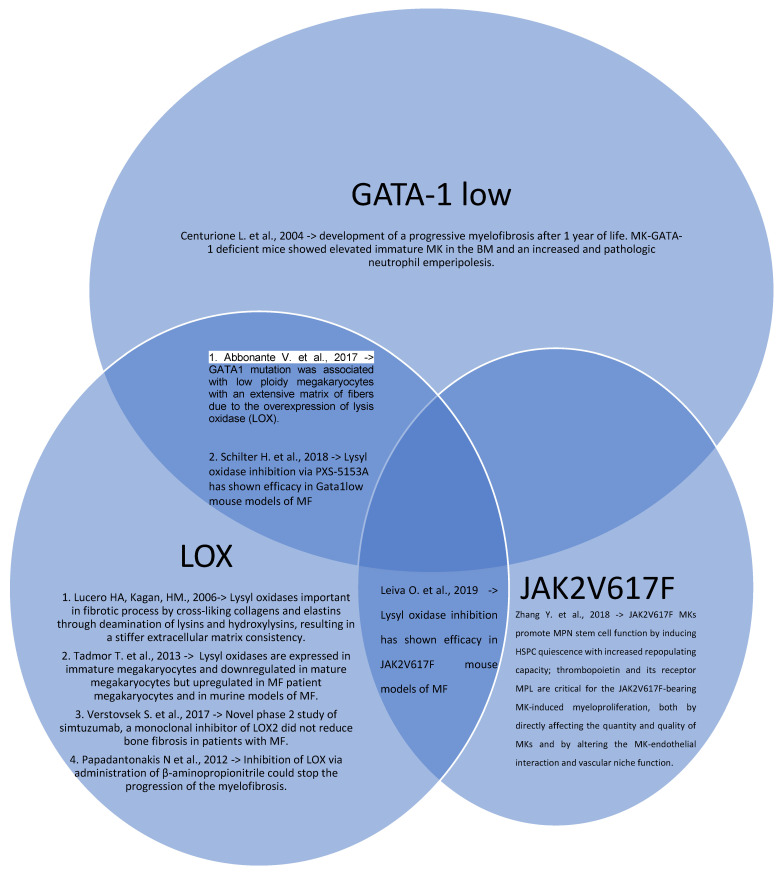
Venn diagram of MPNs in vivo models with their intercorrelations and their main results (Centurione L. et al. [15], Abbonante V. et al. [16], Lucero HA, Kagan, HM. [17], Tadmor T. et al. [18], Schilter H. et al. [19], Leiva O. et al. [20], Verstovsek S. et al. [21], Papadantonakis N et al. [22], Zhang Y. et al. [25]).

### 1.5. In Vitro Models

On the other hand, in vitro cultures of CD34+ hematopoietic stem cells (HSC) from patients with fibrotic MPNs proved that MKs overly expanded, were immature and escaped death signals through anti-apoptotic factor Bcl-xL overexpression [26].

Furthermore, not only MKs from IMF patients secreted higher TGF-β levels than controls, but the ECM microenvironment (especially the fibronectin component) was also able to support progenitor cell proliferation and megakaryopoiesis in a TPO-independent manner [7,11]

In a different aspect, there is evidence from another model that expression of mutant JAK2 in megakaryocytes was sufficient to induce fibrosis and erythropoiesis, the latter due to increased levels of IL6 and other cytokines such as IL-1β [27].

One of the in vitro models of MPNs is based on the purified CD34+ cells from either the bone marrow (BM) or peripheral blood (PB) of patients induced towards megakaryocyte differentiation, while the control is represented by the equivalent cell type from the BM or PB of normal healthy donors, or eventually cord blood cells. A model to replicate the vascular bone marrow niche was represented by endothelial colony forming cells (ECFC), which mimic the vascular niche both in MDS and in MPN. It is important to underline the fact that megakaryocytic maturation and differentiation are regulated by various cytokines, such as IL-6 and TPO, whose release is regulated by normal vascular niche in bone marrow, and cell-surface glycoproteins, such as CD34. Oppositely to this, abnormal ECFC in MDS or MPN express a reduced quantity of these factors, suggesting a possible role of the vascular niche in the maturation and differentiation of hematopoietic cells, in particular those of the megakaryocytic lineage. In particular, it was demonstrated that altered ECFC express less CD34, CD41, AML1, and GPIb, thus impeding the normal megakaryocytic differentiation and/or maturation [28].

Another in vitro study demonstrated that circulating megakaryocytes and platelets from patients with primary myelofibrosis expressed high levels of basic fibroblast growth factor (bFGF). Under culture conditions, bFGF present in megakaryocytes was not exported into the medium, consistent with the fact that bFGF is devoid of a secretion peptide signal. Interestingly, this lack of bFGF secretion was observed in all patients but one, who was in an accelerated phase of the disease and presented an important percentage of circulating megakaryoblasts [29].

Furthermore, Psaila et al. [30] identified and characterized by single-cell RNA sequencing megakaryocyte-biased hematopoiesis in myelofibrosis. Thanks to the single-cell-level resolution of the study, they showed how aberrant megakaryopoiesis in IMF is due to both aberrant differentiation of hematopoietic stem and progenitor cells (HSPCs), as well as proliferation of mature megakaryocytes.

Moreover, they demonstrated that megakaryocytes from IMF patients harbor aberrant metabolic and inflammatory signatures and some aberrant surface markers expression, in particular G6B, an immunoreceptor exclusively found on mature megakaryocytes (Coxon et al., 2017; Senis et al., 2007) [31,32]. G6b-B is a megakaryocyte lineage-specific immunoreceptor tyrosine-based inhibition motif receptor, essential for platelet homeostasis. Mice with a genomic deletion of the entire MPIG6B locus develop severe macrothrombocytopenia and myelofibrosis, which has a human homologous as null mutations in MPIG6B. The current model proposes that megakaryocytes lacking G6b-B develop normally, while proplatelet release is hindered, but the underlying molecular mechanism remains unclear. The mutation of MPIG6B gene is based on a single nucleotide exchange, representing an ideal murine model to study the role of G6b-B. Megakaryocytes from these mice were smaller in size, displayed a less developed demarcation membrane system and reduced expression of receptors. Furthermore, RNA sequencing proved an overall reduction of megakaryocyte-specific transcripts, as well as decreased protein levels of GATA-1, and impaired thrombopoietin signaling. The reduced number of mature MKs in the bone marrow was corroborated on a newly developed MPIG6B null mouse strain. Increased neutrophil emperipolesis into mutant MKs in situ by transmission electron microscopy (TEM) and in cryosections was also observed [33]. In addition to this, it was demonstrated that a small subset of myelofibrosis-associated megakaryocytes expressed an inflammatory cytokine pattern similar to that of normal megakaryocytes, while the majority of myelofibrosis patients had a particular gene expression profile with overall upregulation of profibrotic genes normally expressed at low levels. This finding suggests that megakaryocyte-induced fibrosis in myelofibrosis is due to both expansion of a population of megakaryocytes analogous to those in normal bone marrow as well as the generation of an aberrant population [31,32]. The inflammatory phenotype of megakaryocytes has a solid molecular base: emperipolesis is the main feature that indicates an inflammatory activation of megakaryocytes. Furthermore, emperipolesis is involved in a cascade pathway that promotes even a greater megakaryocytic activation via the releasing of fibroblast growth factor (FGF) and other pro-fibrotic molecules.

**Table 2 ijms-23-07872-t002:** Summary of MPNs in vitro models and their main outcomes.

List of In Vitro Models	Outcomes
Larocca L.M.; Heller P.G.; Podda G. et al. [26] “Megakaryocytic emperipolesis and platelet function abnormalities in five patients with gray platelet syndrome.”	Cultures of CD34+ HSCs from patients with fibrotic MPNs proved that MKs overly expanded, were immature and escaped death signal.
Martyré, M.C. et al. [27] “Elevated levels of basic fibroblast growth factor in megakaryocytes and platelets from patients with idiopathic myelofibrosis.”	Expression of mutant JAK2 in megakaryocytes was sufficient to induce fibrosis and erythropoiesis, the latter due to increased levels of IL6 and other cytokines such as IL-1β.
Teofili L. et al. [28] “Endothelial progenitor cell dysfunction in myelodysplastic syndromes: possible contribution of a defective vascular niche to myelodysplasia.”	Mimicking of the MDS and MPN vascular via ECFC, which express less CD34, CD41, AML1 and GPIb, thus impeding the normal megakaryocytic differentiation and/or maturation.
Villeval J.L.; Cohen-Solal K.; Tulliez M. et al. [29] “High thrombopoietin production by hematopoietic cells induces a fatal myeloproliferative syndrome in mice.”	In vitro cultures with basic fibroblast growth factor (bFGF) present in megakaryocytes showed that it was not exported into the medium, consistent with the fact that bFGF is devoid of a secretion peptide signal.
Psaila, Bethan et al. [30] “Single-Cell Analyses Reveal Megakaryocyte-Biased Hematopoiesis in Myelofibrosis and Identify Mutant Clone-Specific Targets.”	Single-cell RNA sequencing megakaryocyte-biased hematopoiesis in myelofibrosis showed that aberrant megakaryopoiesis in IMF is due to both aberrant differentiation of HSPCs as well as proliferation of mature megakaryocytes.
Coxon C.H.; Geer M.J.; Senis Y.A. [31] “ITIM receptors: more than just inhibitors of platelet activation.”	MK from IMF patients aberrant metabolic and inflammatory signatures.
Senis Y.A.; Tomlinson M.G.; García A.; Dumon S.; Heath V.L.; Herbert J.; Cobbold S.P.; Spalton J.C.; Ayman S.; Antrobus R. [32] “A comprehensive proteomics and genomics analysis reveals novel transmembrane proteins in human platelets and mouse megakaryocytes including G6b-B, a novel immunoreceptor tyrosine-based inhibitory motif protein.”	MK from IMF patients harbor some aberrant surface markers expression, in particular G6B, an immunoreceptor exclusively found on mature MKs.
Becker, Isabelle C. et al. [33] “G6b-B regulates an essential step in megakaryocyte maturation.”	MPIG6B-mutated were smaller in size, displayed a less-developed demarcation membrane system and reduced expression of receptors. RNA sequencing proved an overall reduction of megakaryocyte-specific transcripts, as well as decreased protein levels of GATA-1, and impaired thrombopoietin signaling. Increased neutrophil emperipolesis into mutant MKs in situ by transmission electron microscopy (TEM) and in cryosections was also observed.

### 1.6. Therapeutic Agents

These findings could pave the way to novel therapeutical approaches to MPNs, exploiting immunotherapeutic targeting of stem cell (e.g., CD34) and megakaryocyte (e.g., G6B) surface antigens, with bi-specific antibodies to selectively ‘‘turn off’’ the IMF clone.

In fact, at present, the currently available treatments approved for use in IMF (Ruxolitinib or Fedratinib, JAK inhibitors) [34,35], despite providing symptomatic relief and prolonged survival, do not reduce mutant allele burden or alter the natural history of disease also including the bone marrow failure [36,37], in addition to precarious therapeutic compliance and sustainability due to side effects. There are still limitations in targeting JAK2 that are mainly caused by the dependency of normal hematopoiesis on JAK2, resulting in a specific toxicity expressed as the combination of anemia and thrombocytopenia in patients with MF treated with JAK2 inhibitors [34]. Fedratinib is a selective JAK2-kinase inhibitor which also showed significant reduction in splenomegaly and improvement in constitutional symptoms in patients with MF and was approved in the U.S.; as both a first- and second-line therapy in MF for naïve patients and those for which ruxolitinib therapy failed [38]. There are several other JAK inhibitors that are currently in late phase clinical trials (e.g., momelotinib, pacritinib) and will possibly be adopted in the future therapy of MF. However, other members of the JAK family could also be targetable, such as JAK1, whose signal transduction mediates the dysregulation of cytokines involved in inflammatory processes [39] via its association with JAK2 or JAK3, affecting also the megakaryogenesis [40]. On this basis, Mascarenhas et al. in 2017 focused on the targeting of JAK1 instead of JAK2 using a specific inhibitor, INCB039110, which showed an in vitro low affinity with JAK2 and JAK3 [41]. They showed that the selective inhibition of JAK1 can reduce myelofibrosis-related symptoms without an important hematologic toxicity.

Nonetheless, the only curative therapy is still represented by bone marrow transplantation, which, however, is performed only on a very limited number of patients because it is a complex procedure and burdened with considerable health risks, especially in older patients, and with high costs for the public health system. For this reason, several efforts have been made to understand disease pathology with the ultimate aim of discovering novel therapeutic targets. Back in 2012, Wen Q.J. et al. identified a panel of small molecules which induced MK polyploidization, differentiation and subsequent apoptosis in acute megakaryocytic leukemia (AMKL); among those, MLN8237 (Alisertib), a selective inhibitor of Aurora A kinase (AURKA), proved capable of making AMKL blasts express mature MK markers and displayed potent anti-AMKL activity in vivo [42].

Three years later, the same group studied AURKA inhibition for targeting MK-induced fibrosis in MPNs [43]. After reporting that MKs displayed impaired differentiation also in IMF, they showed that AURKA activity was markedly upregulated in cells with JAK2/CALR activating mutations (probably due to increased c-MYC expression downstream of activated JAK/STAT pathway) [43,44].

In this context, activated megakaryocytes were dysmorphic, grouped themselves into clusters and tended to lose their maturation; therefore, the blockage of AURKA pathway, and consequently the interruption of fibrotic evolution with the stoppage of its main key-player, the megakaryocyte, could represent a future therapeutic strategy to be investigated, so that the ameliorating clinical outcome can align with a bettered morphologic datum, since megakaryocytes tend to normalize and mature after this treatment [43].

Accordingly, they demonstrated that MLN8237 induced maturation, reduced the burden of immature MKs and ameliorated the characteristics of IMF (including bone marrow fibrosis) in JAK2V617F knock-in mice. However, the molecular mechanisms underlying the action of this inhibitor are not known to date. The authors, nevertheless, acknowledged that, even though AUKRA represents a bona fide target in IMF, managing MLN8237 in vivo looks challenging at present (mainly due to its narrow therapeutic window), whereby a further risk stratification for MF-progression is needed to promptly sort patients who would most benefit from anti-fibrotic treatment [43]. Furthermore, according to the results of a multicentric first phase trial conducted by Gangat et al. [45], the AUKRA inhibitor Alisertib was used in 2019 for 24 different patients with myelofibrosis: not only did it reduce splenomegaly and symptomatic burden in circa 1/3rd of the cohort, but megakaryocytes were also normalized and a decrease in fibrosis grading and mutant allele burden in a small subset of patients was also observed. In this therapeutical scenario, Lima K. et al. in 2019 used reversine (a multikinase inhibitor) in MPNs, targeting the SET-2 megakaryoblastic cell line of JAK2 V617F-positive patients, showing a decrease in the activity of AURKA and AURKB and in the expression of antiapoptotic genes, while it promoted a significant increase in the expression of pro-apoptotic genes [46]. The clinical and pathological outcomes were evaluated in a phase II study that involved 87 patients with myelofibrosis regardless of the JAK2 mutational status, showing a significant improvement in ≥50% of patients, while amelioration of splenomegaly was less effective compared to the treatment with ruxolitinib [34]. Another research group hypothesized that TGF beta could be another important therapeutical target in myelofibrosis, even though it was analyzed only in a preclinical setting [47] Bomedemstat (IMG-7289), an inhibitor of LSD1, an enzyme essential for platelet formation, was recently designated for the treatment of ET patients (NCT04254978) [48]. In murine models of MPN, IMG-7289 has shown efficacy in reducing inflammation, fibrosis and other symptomatic criteria, including splenomegaly, in addition to prolonged survival [49]. IMG-7289 targeted Jak2V617F-mutant cells selectively and synergized with Jak inhibition in preclinical MPN mouse models. Bomedemstat is currently in phase IIb clinical trials for patients with myelofibrosis.

Inflammation plays a role in all MPN subgroups, mostly in MF patients. It has been evidenced that inhibiting some cytokines, in particular IL-1β or the NfkB pathway, can either decrease hematopoietic cell growth ex vivo or even diminish fibrosis in vivo [50] Targeting soluble mediators in patients with myelofibrosis is useful mostly to obtain better outcomes concerning constitutional symptoms and reduce frequent comorbid conditions, such as MF-associated anemia. In patients with MF, the reduction of pro-inflammatory cytokines induced by treatment with the JAK1/2 inhibitors correlated with symptomatic improvement [51]. More recently, a research group used mass cytometry and found a limited effect on the levels of pro-inflammatory cytokines in MF patients treated with ruxolitinib with plasma cytokine levels remaining markedly abnormal despite JAK2 inhibition [52]. Some of the elevated cytokines were responsive to pharmacological inhibition of the NfkB and/or the MAP kinase signaling pathway [52], underlining the importance of these pathways for future cytokine-directed therapies in MF and in particular for MF and M-ACT.

Therefore, conjugating molecular profiling with timely clinical advantage is one of the main challenges in the transition from MPN models to MPN patients, as it postulates the need for case-selection at diagnosis [53].

## 2. Conclusions

MPNs represent a fast-evolving field in basic oncohematological research, albeit an ever more mazy puzzle for therapy: amongst sundry potential molecular targets, only a few are selected for translational studies and even fewer are proposed for clinical trials [5]. On the other hand, the latest evidence underlines how indiscriminately restricting MPN management to symptomatic relief and survival gain alone does not interrupt the natural course of the disease, which, conversely, tends to be disabling, irreversible and with highly fatal destination, such as bone marrow failure [5]. At present, the only curative therapy for this event is still represented by hematopoietic stem cell transplantation. Therefore, the transition from MPN models to MPN patients can only be achieved through the evaluation of the morphological data, represented by M-ACT and MKF, whose presence is an index of a faster fibrotic evolution of MPNs and, therefore, can allow a selection of patients to be assigned to a closer follow-up and a more timely and aggressive treatment. In this regard, analyzing the different molecular mechanisms involved in M-ACT and MKF could represent a future perspective to shed light on our comprehension of fibrotic progression in MPNs. Understanding the molecular steps leading to M-ACT and MKF in MPNs would be crucial to identify actionable targets and to develop innovative treatments. Since in this group of hematologic malignancies there are no effective medical treatments, targeting M-ACT and MKF would be crucial. Therefore, our results could represent a starting point for further studies with a larger cohort of patients.

## Figures and Tables

**Table 1 ijms-23-07872-t001:** List of prognostic scores of MPNs from the oldest to the most recent ones and with their respective genetic and/or clinical variables, the subclassification in risk groups and the respective median survival.

Prognostic Model and Risk Factors (Weight)		Risk Groups and Median Survival
**IPSS**		
Hemoglobin < 10 g/dL (1 point)		Low risk: 0 point (135 months)
Leukocytes > 25 × 10^9^/L (1 point)		Intermediate risk-1:1 point (95 months)
Age > 65 years (1 point)		Intermediate risk-2:2 points (48 months)
Circulating blast ≥ 1% (1 point)		High risk: ≥3 points (27 months)
Constitutional symptoms (1 point)		
**DIPSS**. Same variables as IPSS, apart from:		
Hemoglobin < 10 g/dL (2 points)		
		Low risk: 0 point (not reached)
		Intermediate risk-1:1–2 points (14.2 yrs)
		Intermediate risk-2:3–4 points (4 yrs)
		High risk: 5–6 points (1.5 yrs)
**DIPSS+**. Same variables of DIPSS, apart from:		
Unfavorable karyotype (1 point)		Low risk: 0 point (185 months)
Red cell transfusion need (1 point)		Intermediate risk-1:1 point (78 months)
Hemoglobin < 10 g/dL (1 point)		Intermediate risk-2:2–3 points (35 months)
Platelet < 100 × 10^9^/L (1 point)		High risk: 4–6 points (16 months)
**Prognostic model and risk factors (weight)**		**Risk groups and median survival**
**MIPSS70**. Same variables as DIPSS+, apart from:		
Genetic variables	Clinical variables	
One high molecular risk (HMR) mutation (1 point)	Marrow fibrosis grade ≥ 2 (1 point)	Low risk: 0–1 point (not reached)
≥2 HMR mutations (2 points)	Leukocytes > 25 × 10^9^/L (2 points)	Intermediate risk: 2–4 (6.3 yr)
Type 1/like CALR absent (1 point)	Platelet < 100 × 10^9^/L (2 points)	High risk: ≥5 (3.1 yr)
	Circulating blast ≥ 2% (1 point)	
**MIPSS70+ version 2.0**		
Genetic variables	Clinical variables	
VHR karyotype (4 points)	Severe anemia (2 points)	Very low risk: 0 point (not reached)
Unfavorable karyotype (3 points)	Moderate anemia (1 point)	Low risk: 1–2 (16.4 yr)
≥2 HMR mutations (3 points)	Circulating blasts ≥ 2% (1 point)	Intermediate-1 risk: 3–4 (7.7 yr)
One HMR mutation (2 points)	Constitutional symptoms (2 points)	High risk: 5–8 (4.1 yr)
Type 1/like CALR absent (2 points)		Very high risk: ≥9 (1.8 yr)
**GIPSS**. Based on a genetic-only risk factors model.		
VHR karyotype (2 points)		Low risk: 0 point (26.4 yr)
Unfavorable karyotype (1 point)		Intermediate-1 risk: 1 point (8 yr)
Type 1/like CALR absent (1 point)		Intermediate-2 risk: 2 points (4.2 yr)
ASXL1 mutation (1 point)		High risk: ≥3 points (2 yr)
SRSF2 mutation (1 point)		
U2AF1^Q157^ mutation (1 point)		

Specific abnormalities in the megakaryocyte seem to play a central role in the bone marrow fibrotic evolution but also in the induction of myeloproliferation [4,11,12,13].

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
