# Peer review of "A Novel Morphological Parameter Predicting Fibrotic Evolution in Myeloproliferative Neoplasms: New Evidence and Molecular Insights"

_ijms, 2022, doi:10.3390/ijms23147872_

Round 1

Reviewer 1 Report

This review of the prognostic features driving fibrotic progression in MPNs considers certain key molecular aspects of MPN initiation and progression, with particular focus on the role of aberrant megakaryocyte populations and their precursors. An overview of the more important in vitro and in vivo models of MPN is then presented, along with their key findings, before consideration of future therapeutic targets. The authors advance a case for applying their recent description of certain morphological features linked to fibrotic progression in MPN (M-ACT and MKF) as means for furthering insights into myelofibrosis.

The overall structure of the article is sound and the Tables presented are helpful and easy to read (? Table 2 should in fact be a Figure). Figure 1 does not seem to significantly contribute to the article, but rather seems only to reference the author’s own work – the assertion that the megakaryocytes in these images are responsible for the reticulin deposition in these images is questionable. The citations are appropriate and recent. While relevant to the field, there is little novel content in this review with the exception of drawing attention to the authors’ recent publication describing morphological features predicting fibrotic progression in MPN. The link between the author’s description of M-ACT and MKF using histological sections and the main content of the review is rather weak. Moreover, the authors offer no insight into their assertion that their own work can inform and advance the field of fibrotic progression in MPN.

There is repeated and mistaken reference to ‘secretion’ of CD34, CD41 and other surface markers or transcription factors by ECFC. This is an important error and need correction. The text will benefit from extensive editing to improve the English.

The therapeutic agents section is detailed but rather muddled when considering JAK1 inhibitors and Alisertib. These should be considered separately.  

Author Response

We really thank  Reviewer 1 for having carefully analyzed our work and for the detailed suggestions and indications made.

Following the reviewer’s suggestion, Table 2 becomes Figure 1. In addition, following the reviewer’s observation, we have removed  Figure 1 since it does not seem to significantly contribute to the article.

Moreover, regarding the observation that we offer no insight into the role of our work in the field of fibrotic progression in MPN, we want to underline that our center (Fondazione Policlinico Universitario A. Gemelli, IRCCS) represents a reference point for the treatment of MPNs patients in Italy and over the last 5 years we had an average of about 150-170 new diagnoses of MPNs per year. In the last few years, we therefore evaluated the presence of MKF in a large cohort of MPN patients and our preliminary data show that MKF is significantly associated with IMF compared to non IMF MPNs, and this would be of pivotal importance, together with M-ACT, to address the differential diagnosis between IMF and non IMF MPNs. We have also found that the presence of MKF is associated with a faster fibrotic evolution of MPNs and the evaluation of such parameter is easily executable with a high agreement index between pathologists. Although the lack of an independent cohort may represent a limitation to our results, we would like to highlight two issues:

  1. This is the second attempt (after the recent description of M-ACT) to focus on a predictive histo-morphological parameter for MPNs;
  2. Over and above its novelty, our statements are based on the analysis of a wide case population (collected, over a congruous number of years, in one of the most productive Italian MPN centers, which hosts a group known to have extensively published on this subject over the past years), thus concurring to strengthen our observations and to greatly reduce the risk of statistical confounding.

In this regard, analyzing the different molecular mechanisms involved in M-ACT and MKF could represent a future perspective to shed light on our comprehension of fibrosis progression in MPNs. Understanding the molecular steps leading to M-ACT and MKF in MPNs would be crucial to identify actionable targets and to develop innovative treatments.

Following the reviewer’s suggestion, we have corrected the mistaken reference to ‘secretion’ of CD34, CD41 and other surface markers or transcription factors by ECFC. Also, the manuscript was extensively edited by a native English-speaking colleague.

Lastly, following the reviewer’s indication, we considered JAK1 inhibitors and Alisertib separately in the therapeutic agents section and we added a new reference (ref. 44: “ Wen Q, Goldenson B, Silver SJ et al.. Identification of regulators of polyploidization presents therapeutic targets for treatment of AMKL. Cell. 2012 Aug 3;150(3):575-89. doi: 10.1016/j.cell.2012.06.032. PMID: 22863010; PMCID: PMC3613864”).

Reviewer 2 Report

The manuscript by Fiorentino et al. summarizes the current knowledge of MPNs’ prognostic factors related to the development of bone marrow fibrosis with an emphasis on megakaryocytic activation.

In table 1, SRSF2 and U2AF1Q157 mutation is mentioned, but no explanation/description.

In table 2 and 3, for easier finding, the reference number should be added after the first author’s name.

Legend for figure 1 is not clear. What do the authors mean by ”Representative sections in H/E and reticulum” ?

There are several typos which should be corrected.

Author Response

We really thank the Reviewer 2 for the suggestions and indications made.

As suggested by the reviewer, an explanation of SRSF2 and U2AF1Q157 mutations is mentioned in the text: we added in the introduction a brief overview of MPNs’ mutations other than driver mutations and, in particular, we explained that the abovementioned genes regard RNA splicing.

Moreover, based on indications by Reviewer 1, we have removed Figure 1 since it does not seem to significantly contribute to the article, and we also had to turn table 2 into a figure, becoming new Figure 1 (this change was also made on the advice of Reviewer 1). Therefore, the required addition of the reference number after the first author’s name has been made in the new Figure 1 and in Table 2.

Lastly, the manuscript was extensively edited by a native English-speaking colleague and the typos have been corrected.

Round 2

Reviewer 1 Report

The author's have revised their manuscript in line with the reviewer's suggestions and the overall layout is now improved.

While I do have some reservations about the claims made about the robustness of the morphological features described in the author's previous work relating to M-ACT and MKF, it would be unreasonable to include them here as their published data is not the subject of this review. The authors recognise the need for further validation of their findings.

At line 125 it would be helpful if the authors could clarify if they really mean collagen or reticulin fibers.

There remain several minor typographical and grammatical errors throughout the piece that will require editorial input; one example is the grammatical structure of line 124 ending in 'tattoed'. Line 101 also requires correction of 'TGF-'.

Otherwise, I happy to advise acceptance of the article.